# Go Wider: An Efficient Neural Network for Point Cloud Analysis via Group Convolutions

**Can Chen, Luca Zanotti Fragonara ***  **and Antonios Tsourdos**

School of Aerospace, Transport and Manufacturing, Cranfield University, Cranfield MK43 0AL, UK;
can.chen@cranfield.ac.uk (C.C.); a.tsourdos@cranfield.ac.uk (A.T.)

*** Correspondence: l.zanottifragonara@cranfield.ac.uk

**Abstract:** In order to achieve a better performance for point cloud analysis, many researchers apply deep neural networks using stacked Multi-Layer-Perceptron (MLP) convolutions over an irregular point cloud. However, applying these dense MLP convolutions over a large amount of points (e.g., autonomous driving application) leads to limitations due to the computation and memory capabilities. To achieve higher performances but decrease the computational complexity, we propose a deep-wide neural network, named ShufflePointNet, which can exploit fine-grained local features, but also reduce redundancies using group convolution and channel shuffle operation. Unlike conventional operations that directly apply MLPs on the high-dimensional features of a point cloud, our model goes "wider" by splitting features into groups with smaller depth in advance, having the respective MLP computations applied only to a single group, which can significantly reduce complexity and computation. At the same time, we allow communication between groups by shuffling the feature channel to capture fine-grained features. We further discuss the multi-branch method for wider neural networks being also beneficial to feature extraction for point clouds. We present extensive experiments for shape classification tasks on a ModelNet40 dataset and semantic segmentation task on large scale datasets ShapeNet part, S3DIS and KITTI. Finally, we carry out an ablation study and compare our model to other state-of-the-art algorithms to show its efficiency in terms of complexity and accuracy.

**Keywords:** point cloud; semantic segmentation; shape classification; feature shuffle; neural network

## 1. Introduction

Point cloud processing is an increasingly important task for a wide range of applications such as: environment perception for autonomous driving [1–4], virtual reconstruction [5], or augmented reality (AR) [6]. It belongs to a deep learning branch that is typically referred to as *geometric deep learning* [7]. However, it is still challenging to analyse and extract useful features of the underlying shape representation efficiently because of the disadvantages of having a large amount of points and an unstructured distribution. On the other hand, point cloud data can provide a very accurate geometric information of a three-dimensional object.

Considering the remarkable success and advantages of Convolutional Neural Networks (CNNs), various authors firstly tried to apply them over standard grid structure voxelized from an unordered point cloud, as CNNs only work on a regular grid data structure [8,9]. However, this intuitive way of applying deep learning to point clouds leads to high memory and computation requirements due to the natural sparse and irregular structure of the data. *PointNet* [10] treats point cloud directly as a set of unordered points, leveraging a Multi-Layer-Perceptron (MLP) network and a symmetric function (e.g., max pooling) to exploit global features and make unordered points invariant to permutations. The main drawback is that the local information is not used in the stacked MLP layers. In order to solve

this issue, *PointNet++* [11] proposes a hierarchical neural network joining *PointNet* with a sampling and grouping layer to capture local representation within the data. Similarly, *DGCNN* [12] extracts local features by introducing an edge convolution operation on points and edges connecting each point and corresponding neighbours. Chen et al. [13] proposes an attention-aware neural network to learn local features by highlighting different attention coefficient for neighbouring points. Another CNN-like approach is *PointCNN* [14], which manages to transform an unordered point cloud to a latent canonical order by using a $\chi$-convolutional operator. *RS-CNN* [15] and *ConvPoint* [16] attempt to learn irregular CNN-like filters to capture local features for point cloud. Our model applies feature shuffling operations on *PointNet++* [11] for less model computation but still with high accuracy performance. As can be seen in Figure 1, it performs the best trade-off in accuracy and model complexity.

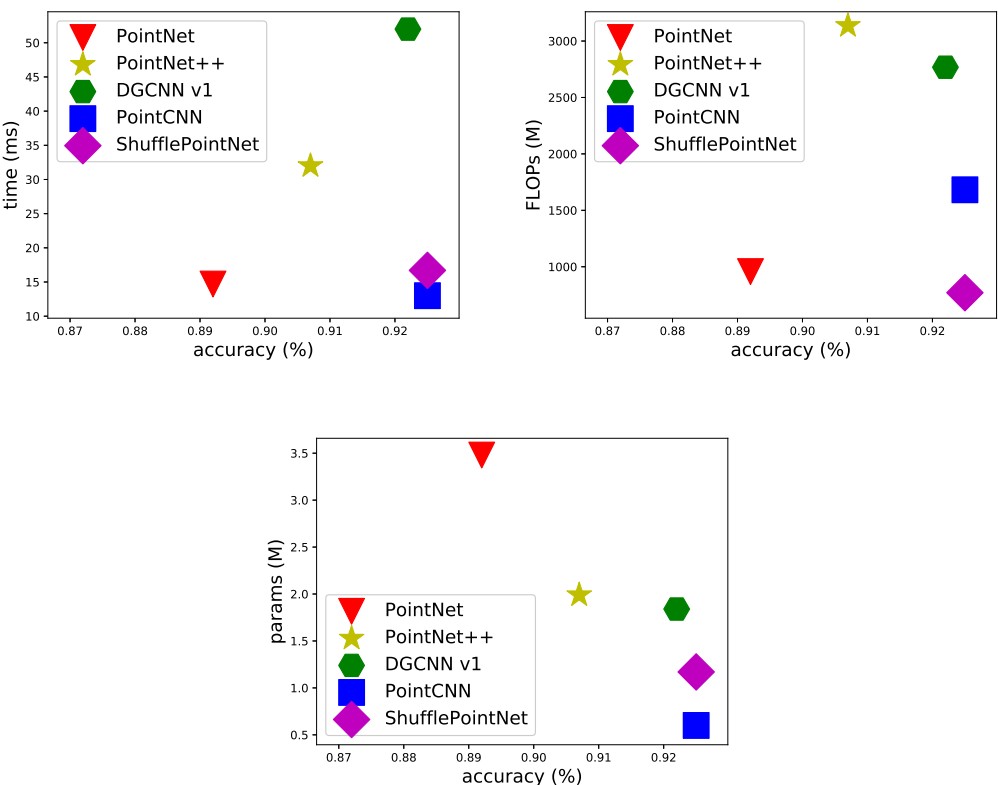

**Figure 1.** Measurement of accuracy, parameters, floating point operations (FLOPs) and forward time. It shows that our proposed ShufflePointNet (magenta diamond marker) tends to stay in the bottom-right corner, which means that our model conjugates high accuracy with low model complexity.

We notice that the modern neural networks proposed for geometric deep learning [10–13] heavily relies on dense MLP used to build repeated block structures with a different number of filters. On the other hand, the elimination of redundancies within these dense MLP operations is rarely mentioned for point cloud analysis, although it is a relatively common topic in the computer vision domain [17–21]. Within the point cloud processing domain, many applications (e.g., autonomous driving) normally have to process large scale point clouds, which make model complexity increase significantly. Besides this, the MLP operation is not a spatial convolution, which limits the local feature extraction capabilities of the overall model. As a result, applying a large number of deep MLP convolutions easily leads to data overfitting. However, the MLP operation is efficient in dealing with unstructured data (e.g., point cloud, social networks) as regular CNNs are only suited to deal with data having a standard grid structure. On the other hand, these dense and deep $1 \times 1$ convolutions are not efficient in terms of computation and memory. Therefore, we draw attention to the use of sparse MLP

convolutions over irregular point clouds, which can leverage the advantages of MLP convolutions but also reduce the redundancies by making use of sparse MLP convolutions.

In the area of point cloud analysis, there are also other methods proposed to decrease model complexity. For instance, the models proposed in [14,15] attempt to develop efficient CNN-like kernels to reduce the use of MLP-like operations. Conversely, *PointNet++* [11] leverages the use of Farthest Point Sampling (FPS) to downsample the points in advance before applying the PointNet architecture to the features. However, only few researchers studied the impact of channel extension and wider neural networks. Inspired by group convolutions as introduced by *AlexNet* and *ShuffleNet* [20,22], we primarily focus on building a deep-wide neural network to reduce redundancy and also achieve high performance. The main contributions of this paper are summarized as follows:

- We propose an efficient deep-wide neural network, named *ShufflePointNet*, aiming at significantly reducing redundancies in depth and exploiting more features information in width for better point cloud understanding.
- We split features into groups and shuffle them to allow features between groups to exchange respective information.
- We evaluate our model on both small point cloud dataset (e.g., ModelNet40) and real world large-scale point cloud datasets (e.g., S3DIS and KITTI), and we compare both accuracy and latency performances to various state-of-the-art models.
- To the best of our knowledge, ShufflePointNet is the first deep-wide model that employs the channel shuffling operation and group convolution to efficiently capture local representations for point cloud.

## 2. Related Work

### 2.1. Volumetric Grid and Multi-View Methods

Volumetric methods [8,23–25] convert irregular point cloud to regular dense grid and voxels to allow feature extraction by standard CNNs. However, applying CNNs over dense volumetric grid leads to extremely high costs for computation and memory. Deep *kd-net* [23] and *octnet* [25] improve efficiency in space partition and resolution, but still cause loss of local geometric information due to bounding voxels.

Multi-view methods [26,27] apply classic 2D CNNs on groups of 2D images obtained from different views of 3D objects. However, the 3D geometric shape is unlikely to be captured from these 2D images due to lack of depth information. In addition, part of points information might be lost due to the occlusions on the images. As a result, it is non-trivial to segment every point for classification.

### 2.2. Deep Learning Methods Directly on an Unstructured Point Cloud

*PointNet* [10] takes the lead in treating point clouds as a set of unordered points and applying stacked Multi-Layer-Perceptron (MLP) to learn individual point features. The global feature is finally obtained by a symmetric function (e.g., max pooling). However, local features that are beneficial to better understanding a geometric shape are not considered in this approach. In order to improve the performance and overcome this problem, *PointNet++* [11] constructs a hierarchical neural network that applies recursively *PointNet* on the local features that combine the sampled points with the corresponding neighbouring points. *DGCNN* [12] applies *PointNet* on the edge features concatenating each point and its edges connecting the corresponding point and its neighbouring pairs. *PointCNN* [14] transforms a given irregular point cloud to a latent canonical order by learning a $\chi$-convolutional operator, after which classic 2D CNNs are used for local feature extraction. [15] attempts to learn a customized convolutional weight from the geometric local relation for a shape-aware representation.

### 2.3. Geometric Deep Learning Methods

Geometric deep learning [7] is the modern terminology for deep neural network techniques addressing non-Euclidean structured data (e.g., point cloud, social networks). Graph CNNs [28–30] have achieved great success in many tasks for graph representation of non-Euclidean data. *Superpoint* [31] organizes point set into geometric elements, after which a graph CNN structure is applied to exploit the local features.

## 3. Model Structure

In this section, we explain in more detail our model architecture: firstly, we introduce a process of construction of a directed graph for local feature representation as shown in Figure 2. Then we present group convolution and channel shuffle as shown in Figure 3. At last, we explain our MLP Shuffled Group Convolution unit (SGC unit for short) and the overall model architecture in detail (see Figures 4 and 5 respectively).

We define $\mathbf{X} = \{\mathbf{x}_i \in \mathbb{R}^F, \quad i = 1, 2, \dots, N\}$ as a raw point set and input for our model, where $F$ is the dimension of the point-cloud representation, $N$ is the number of points, and $\mathbf{x_i}$ is typically the 3D coordinate of each point in a Cartesian reference system. The dimension $F$ is not necessarily limited to 3 because it is possible to use other features for each point such as: color, intensity, or the normal component with respect to the surface.

### 3.1. Local Feature Representation

A trending way of representing point cloud data is to assume them as graph-like structures with nodes and edges [12], where each node and the corresponding edges on a graph are naturally defined by the data points. As a result, converting a point cloud to a directed acyclic graph and applying neural networks on the graph structure is an efficient way of learning the embedded information for the neighbourhood of each node.

Hence, we construct a directed acyclic graph $G = (V, E)$ for a specific point set (see Figure 2), where $V \subseteq \mathbb{R}^F$ is the node corresponding to each point of the point cloud, and $E$ indicates the edges connecting the central point to its neighbourhood. As shown in Figure 2, assuming that the input of a point set can be represented by an uniform distribution, we choose a *k*-nearest neighbour (*k*-NN) search to explore the neighbourhood of each point, as it can guarantee a fixed number of neighbours. It is important to stress that, for selecting the neighbouring nodes, the geometrical Euclidean distance between the central point and the neighbours is used. We introduce $K$ as one hyper-parameter of our model, which is the number of elements in the neighbourhood of a central point. Once a neighbourhood is defined, we can also introduce an extra feature linked to the graph-like structure, which is the directed edge connecting the central point with the j-th neighbour as $\mathbf{e}_{ij} = (\mathbf{x}_i - \mathbf{x}_{ij})$, where $\mathbf{x_i} \in V$ are the i-th central point features and $\mathbf{x_{ij}}$ are the j-th neighbouring point features with respect to the i-th central point $\mathbf{x_i}$. Hence, we can aggregate this new set of features into an augmented feature vector by simple concatenation: $\hat{\mathbf{x}}_{ij} = \left[\mathbf{x}_i, \mathbf{e}_{ij}\right]^T$, having a feature dimension $\hat{\mathbf{x}}_{ij} \subseteq \mathbb{R}^{2F}$. For simplicity, in the rest of the paper, we are going to keep referring to $F$ as the number of features, but the reader should keep in mind that $F$ has twice the dimensions when taking into consideration the local feature and using an augmented feature space.

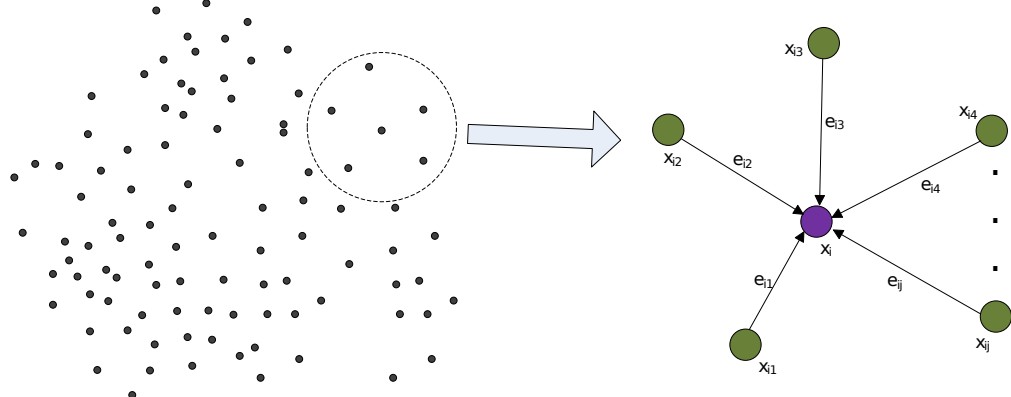

**Figure 2.** The process of construction of a directed acyclic graph for local feature extraction. On the left is the raw point cloud data, on the right, the graph-like structure extracted from part of the point cloud is represented. $\mathbf{x_i}$ indicates the central point, $\mathbf{e_{ij}}$ are the directed edges connecting center point and its corresponding neighbouring points $\mathbf{x_{ij}}$.

### 3.2. Group Convolution

*AlexNet* [22] was the first architecture to introduce group convolutions, mainly aiming at distributing the model training over two GPUs. Its effectiveness is also well demonstrated by [17–21] in the image domain. In the author's understanding, group convolution presents three main advantages. Firstly, it can ease the training of deep neural networks by reducing their redundancy. Secondly, it is an efficient method to increase the width of neural networks allowing more feature channels which is beneficial to encode more information on each group if we consider a constrained complexity budget. Thirdly, it can be also treated as sparse convolution, as convolution kernels on each group are only responsible for part of the entire feature channel. Furthermore, it can also prevent overfitting to some extent.

The memory size and computational complexity (FLOPs) for a single $1 \times 1$ group convolution can be measured by Equations (1) and (2) [32], respectively. It is trivial, but worth highlighting that it becomes a standard MLP operation when we set the number of groups to $g = 1$. Further to this, the number of parameters and FLOPs are reduced to $1/g$ for a single operation when we split $k$-nn graph to $g$ groups:

$$params = 1 \times 1 \times \frac{C_{in}}{g} \times \frac{C_{out}}{g} \times g = \frac{C_{in} \times C_{out}}{g} \tag{1}$$

$$FLOPs = \frac{N \times K \times C_{in} \times C_{out}}{g} \tag{2}$$

where $C_{in}$ and $C_{out}$ indicate the input and output channel, respectively, $N$ is the number of points, $K$ is the number of neighbouring points and $g$ is the aforementioned number of groups.

### 3.3. Channel Shuffle

Considering the fact that each group only holds incomplete and partial representations of the graph features, it is unlikely to capture sufficient features if there is no communication among multiple group convolutions. Hence, inspired by [20], we decided to stack group convolutions together and to shuffle the feature channel in order to achieve a satisfactory fusion of the features for all groups. Specifically, as shown in Figure 3, assume we split the feature channel into two groups (green color on the left and blue color on the right), each of which has three channels. After MLP operation, each group has three filters for simplicity. We interleave every filter in order between two groups, after which the output is shown as the last process in Figure 3.

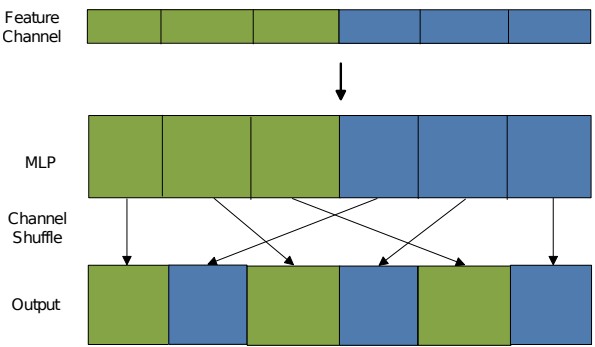

**Figure 3.** Ordered channel shuffle with MLP operation.

### 3.4. Computational Complexity Metrics

In order to prove the efficiency of group convolutions, we have to introduce several metrics aimed at measuring the model complexity, such as: memory size, computational cost and forward time. In more detail, the memory size is defined as the total number of parameters for the convolution kernels, the number of floating point operations (FLOPs) indicates the computational cost of neural networks and, finally, the forward time represents the forward propagation time of a neural network. FLOPs are a widely used but indirect metric of complexity, and it can approximate a model speed by measuring the total number of operations, such as the number of multiply-adds operation, happening in a neural network. The forward propagation time is normally used as direct metric to measure a model speed (particularly useful for real-time applications). It is worth mentioning that FLOPs and forward time also vary in function of the used platform/device. Therefore, it is necessary to align the platform environment when comparing the complexity of different models using these metrics.

### 3.5. MLP Shuffled Group Convolution Unit

As shown in Figure 4, we extract from the point cloud a group of features having dimension $N \times K \times F$ as input, where $N, K, F$ indicate the number of points, neighbours and feature channels, respectively. Assume the dimension of the output feature is $N \times K \times f$. For the standard MLP convolution, we apply $1 \times 1$ convolution with $f$ filters to input features, and the number of parameters is $F \times f$. For our SGC unit, the feature channels is firstly split into $g$ groups, and the feature dimension of each group becomes $N \times K \times \frac{F}{g}$, where $g$ is the number of groups. Secondly, we apply an MLP convolution with filters $\frac{f}{g}$ on each group. We then concatenate all the groups of the feature channels to generate a feature block with dimension $N \times K \times f$. In order to enable communication for all the groups, we finally shuffle the feature channels to fuse the features, which then becomes the input of the next layer. The number of parameters for a SGC unit is finally reduced to $\frac{F \times f}{g}$.

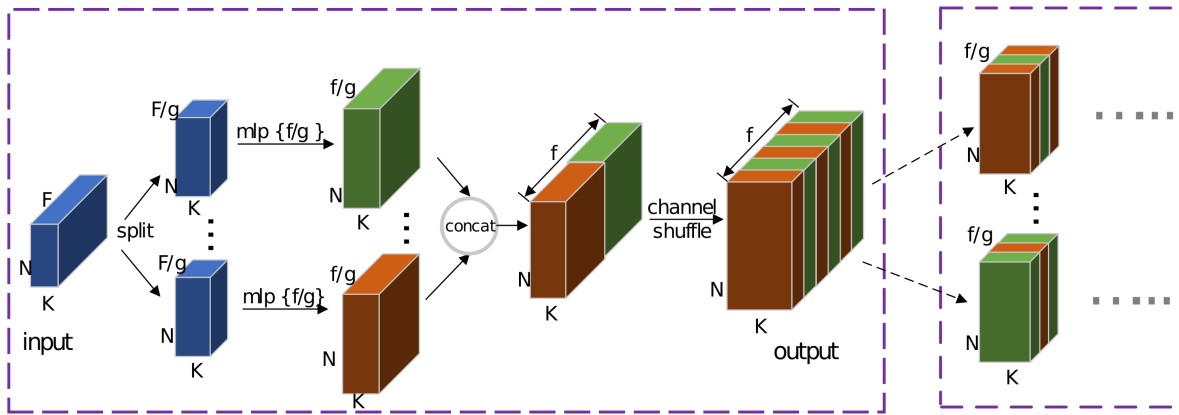

**Figure 4.** MLP Shuffled Group Convolution unit (SGC unit).

*3.6. Model Architecture*

The ShufflePointNet architecture can be used for shape classification and segmentation as shown in Figure 5. We use the *PointNet++* architecture [11] as our backbone. However, there are three main differences between our model and *PointNet++*. Firstly, we use $1 \times 1$ group convolutions to exploit the local, fine-grained features in terms of both depth and width. Secondly, instead of the radial search for neighbouring points used by *PointNet++*, we employ a *k*-NN search to guarantee a fixed number of neighbours. Thirdly, instead of the neighbouring point feature, we use the edge feature connecting each point and the corresponding neighbours to explicitly represent the relative position of a neighbouring point with respect to its center point.

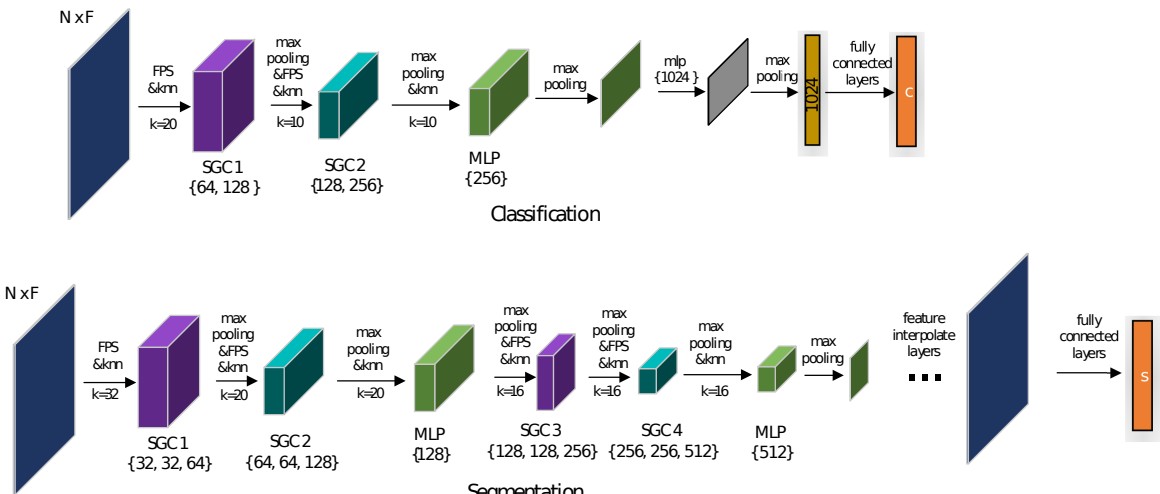

**Figure 5. ShufflePointNet architecture:** The architecture uses *PointNet++* as backbone, and it contains classification structure (top branch) and semantic segmentation structure (bottom branch) for point cloud. In detail, $N, F$ indicates the number of input points, and corresponding feature channels, respectively. $k$ is the number of the neighbours. In addition, the number in the brace {} indicates the MLP filters. The classification model recursively applies Furthest Point Sampling (FPS) and a Shuffled Group Convolution (SGC) module to capture downsampled local features, followed by a shared full-connected layer to generate a global feature, which is used to predict classification scores. The segmentation model extends the classification model by two more SGC modules and a feature interpolation module to obtain a category score for each point.

3.6.1. Classification Model Structure

Our classification model is shown in Figure 5 (top branch). We start by sampling a subset of 512 points from the point cloud by using the farthest point sampling (FPS) method and we also cluster *k* neighbouring points for each central point by the *k*-NN method for extracting the local geometric feature in the following layer of MLP Shuffled Group Convolution (SGC) with filters (64, 128). Similarly, 128 points are then subsampled, followed by an SGC unit with filters (128, 256). The third layer is used to aggregate the local features by using an MLP layer having 256 dimension. For the fourth layer, the global feature of the point cloud is extracted by a max pooling operation. At last, three fully-connected layers (512, 256, 40) are used to transform the global feature to 40 categories. A dropout operation with a keep probability of 0.5 is also used in the last three fully connected layers. ReLU was used as an activation function and batch normalization is used for all the edge convolution operations and the fully-connected layers. In addition, the numbers *k* of nearest neighbours for the first, second and third layer are 20, 10 and 10, respectively.

### 3.6.2. Segmentation Model Structure

Our segmentation model is shown in Figure 5 (bottom branch). We extend the classification model to six layers for local features extraction, which are SGC with filters (32, 32, 64), SGC with filters (64, 64, 128), standard MLP with filters (128), SGC with filters (128, 128, 256), SGC with filters (256, 256, 512) and standard MLP with filters (512), respectively.

In order to generate interpolated points from the given subsampled points and the corresponding features, we use the inverse squared Euclidean distance weighted average function based on each point and the corresponding $k$ nearest neighbours [11] as shown in Equation (3), where $\omega_i(\mathbf{x}) = \frac{1}{(\mathbf{x}-\mathbf{x_i})^2}$ is the inverse square Euclidean distance between $x$ and $x_i$:

$$f(\mathbf{x}) = \frac{\sum_{i=1}^{k} \omega_i(\mathbf{x}) f_i}{\sum_{i=1}^{k} \omega_i(\mathbf{x})} \tag{3}$$

We then concatenate interpolated points features with corresponding abstraction level point features as shown in Figure 6, and apply MLP convolution to fuse them together. Next, four upsampling layers with respective filters (256, 256), (256, 256), (256, 128), (128, 128, 128) are employed to extract fine-grained features for each point. Similar to classification structure, the global feature of the point cloud is obtained by a max pooling; we then duplicate the global feature with $N$ times and finally apply four multiple layer perceptron (MLP) layers (256, 256, 128, $s$) with dropout probability 0.6 to transform the global feature to $s$ categories. The number of the points $N$ is 2048, 4096 and 16,384 for the ShapeNet part dataset [33], the Stanford Large-Scale 3D Indoor Spaces Dataset (S3DIS) [34] and the KITTI dataset [35], respectively. $s$ indicates the number of classes.

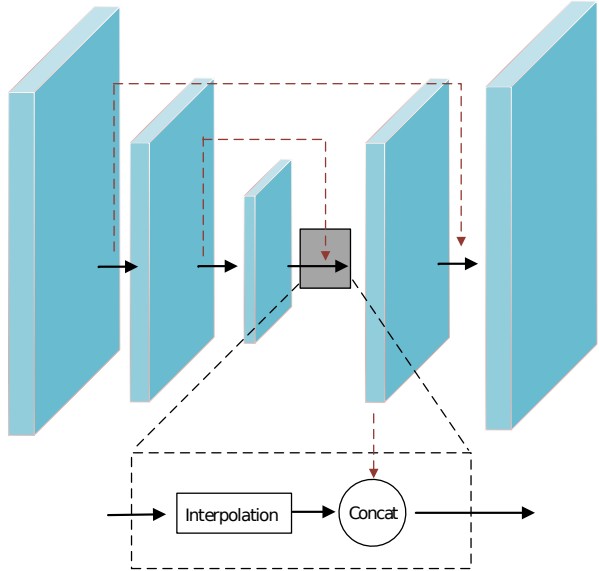

**Figure 6.** Upsampling structure.

## 4. Experiments

In this section, we carry out comprehensive experiments aimed at evaluating the performance of ShufflePointNet when dealing with shape classification and segmentation tasks. In order to demonstrate the effectiveness of our model, we highlight the trade-off between accuracy and complexity of different recent state-of-the-art methods with respect to our model. Finally, we carry out an ablation study to investigate different model settings. We also place implementation issues for training on our Github https://github.com/FrankCAN/ShufflePointNet.

*4.1. Classification*

4.1.1. Dataset

We evaluate our classification model on the ModelNet40 benchmark [36]. It contains 9843 training models and 2468 testing models that are classified to 40 classes. We firstly uniformly downsampled all models to 1024 points from a total of 2048 points, and then normalized them into a sphere with radius one. Besides this, in order to improve the overall robustness, we further augmented the training data by means of rotations, by random scaling the sampled points, and by jittering the position of every point using Gaussian noise with zero mean and 0.01 standard deviation.

4.1.2. Training Details

Our model is implemented using TensorFlow v1.6. We employed Adam as an optimization algorithm [37], with a batch size of 32 samples for training. The learning rate has been initially set to 0.001, and then allowed to decay with a rate of 0.7 every 20 epochs down to 0.00001. The momentum coefficient for batch normalization starts from 0.9 and increases gradually with a growth rate of 0.5 every 20 epochs up to 0.99.

Our classification model was trained for approximate 3 h on a ThinkStation P920 workstation (Getech Ltd., Ipswich, UK) with 12 Intel E5-1650 V4 CPUs, 3.6 GHZ, 32 GB memory, 1 NVIDIA GTX 1080Ti (Novatech Ltd, Portsmouth, UK).

4.1.3. Results

Table 1 compares the performance of ShufflePointNet for classification to recent state-of-the-art models in terms of accuracy and model complexity. Specifically, we decided to use as metric the mean per-class accuracy (mA%), the overall accuracy (OA%), the number of model parameters (Million), FLOPs (Million), and forward time (ms) on the ModelNet40 benchmark dataset [36]. It is worth mentioning that, compared to the backbone structure *PointNet++*, our model outperforms it by 1.4% in terms of accuracy, whilst also significantly reducing the amount of parameters, FLOPs and forward time by 41%, 75% and 47.8%, respectively. These results are strongly supporting the effectiveness of our deep-wide model as a reliable solution having high accuracy and lower computational complexity.

**Table 1.** Classification results on the ModelNet40 dataset.

| Methods | mA(%) | OA(%) | Params | FLOPs | Time |
|---|---|---|---|---|---|
| VoxNet [8] | 83.0 | 85.9 | - | - | - |
| PointNet [10] | 86.0 | 89.2 | 3.48 M | 957 M | 14.7 ms |
| PointNet++ [11] | - | 90.7 | 1.99 M | 3136 M | 32.0 ms |
| KC-Net [38] | - | 91.0 | - | - | - |
| SpecGCN [39] | - | 91.5 | 2.05 M | 1112 M | 11254 ms |
| KD-Net [23] | - | 91.8 | - | - | - |
| DGCNN [12] | **90.2** | 92.2 | 1.84 M | 2768 M | 52.0 ms |
| PCNN [40] | - | 92.3 | - | - | - |
| Ours | 90.1 | 92.5 | 1.17 M | **770 M** | 16.7 ms |
| PointCNN [14] | 88.8 | 92.5 | **0.6 M** | 1682 M | **13.0 ms** |
| RS-CNN [15] | - | **93.6** | - | - | - |

Our model achieves comparable performance with PointCNN [14], which is also point-based-convolution methods over an unordered point cloud. PointCNN attempts to permute the neighbouring points into a canonical order by learning a feature transformer, after which standard CNNs are available to use. However, the performance is far from ideal due to the fact that the positions of neighbours are unordered and unpredictable, while ShufflePointNet uses MLP (Multi-Layer-Perceptron) convolutions over center points and corresponding neighbours, which is

naturally invariant to point cloud ordering. In addition, the deep and wide architecture allows for exploiting sufficient features with less computation. However, the receptive field of CNN is much larger than that of MLP, which leads to the fact that PointCNN still can achieve remarkable performance by leveraging powerful CNNs over approximate ordering of neighbours.

It is also possible to observe that *PointCNN* [14] has larger FLOPs but minor forward time, while *SpecGCN* [39] has similar FLOPs to *PointNet* but requires a much longer inference time. In addition, these results confirm our hypothesis that the use of FLOPs is an indirect method of estimating the model complexity, and it cannot be used as the only metric to estimate the computational efficiency of a deep learning model.

### 4.1.4. Ablation Study

Finally, we tested our classification model using different hyperparameter settings on the ModelNet40 benchmark dataset [36].

Table 2 shows the change in performance for different number of groups. In order to evenly split the feature channels, the initial input of the SGC unit shares the same edge feature $e_{ij} = (x_i, x_i - x_{ij})$ for all groups. The results highlight how the use of multiple groups (e.g., $g = 2, 4$) is beneficial with respect to having no grouping operation (i.e., $g = 1$), as multiple groups help with capturing more information for a given complexity. However, when the group number becomes too large (e.g., $g = 8$), the performance decreases. Hence, it is likely that the usefulness of the feature information becomes limited for each group when we split feature channels into many groups, leading to the fact that the neural network is unlikely to learn useful information from individual groups. The number of parameters and FLOPs only show a slight drop with respect to the number of groups because of Equations (1) and (2). The reason is that standard MLP operations (e.g., global feature extraction) and the fully-connected layers take almost 95% of the computational cost and are not influenced by the group number. The forward time also increases when the group number becomes larger, as more groups need more time to store corresponding features and parameters to the cache [32].

**Table 2.** Model complexity and accuracy for different groups.

| Groups | Shuffling | OA (%) | Params | FLOPs | Time |
|--------|-----------|--------|--------|-------|------|
| $g = 1$ | × | 91.8 | 1.20 M | 946 M | 15.3 ms |
| $g = 2$ | ✓ | 92.5 | 1.17 M | 770 M | 16.7 ms |
| $g = 2$ | × | 91.9 | 1.17 M | 770 M | 16.7 ms |
| $g = 4$ | ✓ | 92.3 | 1.15 M | 692 M | 19.5 ms |
| $g = 8$ | ✓ | 91.5 | 1.14 M | 650 M | 22.7 ms |

As shown in Table 2, when we split feature channels into two groups and disable channel shuffling operation, the performance of overall accuracy drops significantly to 91.9%. We discuss the fact that, when we disable shuffling operation, center point features and corresponding neighbouring points features are captured, respectively, but still will concatenate before the max pooling layer. As a result, without shuffling, only the last MLP layer contributes to exploiting local features from both center points and corresponding neighbouring points.

Table 3 shows different settings for the number of points, low-level edge features $e_{ij}$, and grouping methods. It indicates that there is no improvement when we add neighbour features to represent local features, and edge features connecting center point to corresponding neighbourhood is beneficial for better results. In addition, $k$-NN searching slightly outperforms radius searching. We discuss the reason that $k$-NN searching is more efficient when the layout of dataset can be more or less treated as uniform distribution.

**Table 3.** Ablation study of ShufflePointNet.

| Model | Points | Edge Feature $e_{ij}$ | Grouping | OA (%) |
|-------|--------|----------------------|----------|--------|
| A | 1k | $(\mathbf{x_i}, \mathbf{x_i} - \mathbf{x_{ij}})$ | knn | 92.5 |
| B | 1k | $(\mathbf{x_i}, \mathbf{x_{ij}})$ | knn | 92.0 |
| C | 1k | $(\mathbf{x_i}, \mathbf{x_{ij}}, \mathbf{x_i} - \mathbf{x_{ij}})$ | knn | 92.5 |
| D | 2k | $(\mathbf{x_i}, \mathbf{x_i} - \mathbf{x_{ij}})$ | knn | 92.7 |
| E | 1k | $(\mathbf{x_i}, \mathbf{x_i} - \mathbf{x_{ij}})$ | radius search | 92.1 |

*4.2. Semantic Segmentation*

The segmentation model has been evaluated on the ShapeNet part dataset [33], the Stanford Large-Scale 3D Indoor Spaces Dataset (S3DIS) [34], and the KITTI dataset [35].

4.2.1. ShapeNet Part Dataset

This dataset is composed of 16,881 CAD models (14,007 training models and 2874 testing models) that are classified to 16 categories, and each model is annotated with several parts (less than 6) from 50 part classes. We follow the same sampling strategy as Section 4.1 to sample 2048 points uniformly. The task is to classify each point as a part category from models.

4.2.2. S3DIS Dataset

S3DIS [34] uses a Matterport scanner to collect six-dimensional (XYZ, RGB) point clouds, which are then processed to a nine-dimensional feature space (XYZ, RGB, normalized spatial coordinate), from 271 rooms in six areas. We used the same settings as in the *DGCNN* article [12] to slice all the rooms into 1 m by 1 m blocks, each of which are then downsampled to 4096 points during the training process. At the same time, we used all the points to evaluate our model. We tested our model on area number 5, whilst using the other areas for training.

4.2.3. KITTI Dataset

The KITTI Object Detection Benchmark [35] was used to evaluate our model on a real traffic scene. In this case, we used the same configurations as in [41] to separate the KITTI dataset to 7481 samples for training and 191 samples for testing. However, due to the fact that each frame contains approximately 100,000 points, it is infeasible to apply all the points as input to our model. Therefore, we downsampled the points from about 100 k to 16 k (specifically to 16,384 points). Firstly, we removed points outside the field of view, and then we randomly selected 11,469 points (70%) having a distance of less than 40 m from the sensor and 4915 (30%) points from the remaining data.

4.2.4. Training Details

By using the same training settings as in the classification task, with the exception of using a batch size of 24, the model was trained for approximately 5 h for the ShapeNet part dataset [33] on a cluster server with four Intel E5-2620 v4 CPUs giving 32 CPU cores, 256 GB of shared memory, and two NVIDIA TESLA V100 GPUs.

4.2.5. Results

The mean Intersection over Union (mIoU) [10] is used as a metric to evaluate segmentation performance. The IoU is calculated by averaging the IoUs for all parts belonging to the same categories; then, the mIoU is computed as the mean IoUs for all shapes from the testing dataset.

For the semantic part segmentation task, Table 4 indicates that our segmentation model achieves competitive results on the ShapeNet part dataset [33]. It is the best performing model in four categories, similarly to the *DGCNN* and *SGPN* state-of-the-art models. Figure 7a illustrates some of the segmented

shapes, whilst Figure 7b visualizes the errors of our prediction results compared to the ground truth results. We present the ground truth on the left side and our predictions on the right side of the image.

**Table 4.** Semantic part segmentation results on ShapeNet part dataset.

|  | avg | air. | bag | cap | car | cha. | ear. | gui. | kni. | lam. | lap. | mot. | mug | pis. | roc. | ska. | tab. |
|---|---|---|---|---|---|---|---|---|---|---|---|---|---|---|---|---|---|
| kd-net [23] | 82.3 | 82.3 | 74.6 | 74.3 | 70.3 | 88.6 | 73.5 | 90.2 | 87.2 | 81.0 | 94.9 | 57.4 | 86.7 | 78.1 | 51.8 | 69.9 | 80.3 |
| Kc-Net [38] | 83.7 | 82.8 | 81.5 | 86.4 | 77.6 | 90.3 | 76.8 | 91.0 | 87.2 | 84.5 | 95.5 | 69.2 | 94.4 | 81.6 | 60.1 | 75.2 | 81.3 |
| PointNet [10] | 83.7 | 83.4 | 78.7 | 82.5 | 74.9 | 89.6 | 73.0 | **91.5** | 85.9 | 80.8 | 95.3 | 65.2 | 93.0 | 81.2 | 57.9 | 72.8 | 80.6 |
| 3DmFV [42] | 84.3 | 82.0 | 84.3 | 86.0 | 76.9 | 89.9 | 73.9 | 90.8 | 85.7 | 82.6 | 95.2 | 66.0 | 94.0 | 82.6 | 51.5 | 73.5 | 81.8 |
| RSNet [43] | 84.9 | 82.7 | **86.4** | 84.1 | **78.2** | 90.4 | 69.3 | 91.4 | 87.0 | 83.5 | 95.4 | 66.0 | 92.6 | 81.8 | 56.1 | 75.8 | 82.2 |
| PointNet++ [11] | 85.1 | 82.4 | 79.0 | **87.7** | 77.3 | 90.8 | 71.8 | 91.0 | 85.9 | 83.7 | 95.3 | 71.6 | 94.1 | 81.3 | 58.7 | 76.4 | 82.6 |
| DGCNN [12] | 85.1 | **84.2** | 83.7 | 84.4 | 77.1 | **90.9** | 78.5 | **91.5** | 87.3 | 82.9 | **96.0** | 67.8 | 93.3 | 82.6 | 59.7 | 75.5 | 82.0 |
| SGPN | **85.8** | 80.4 | 78.6 | 78.8 | 71.5 | 88.6 | 78.0 | 90.9 | 83.0 | 78.8 | 95.8 | **77.8** | 93.8 | **87.4** | 60.1 | **92.3** | 89.4 |
| OURS | 85.1 | 82.7 | 83.6 | 86.2 | 77.3 | 90.3 | **78.7** | 90.9 | **87.4** | 84.6 | 94.6 | 68.5 | **94.5** | 82.9 | 51.3 | 73.4 | 82.3 |

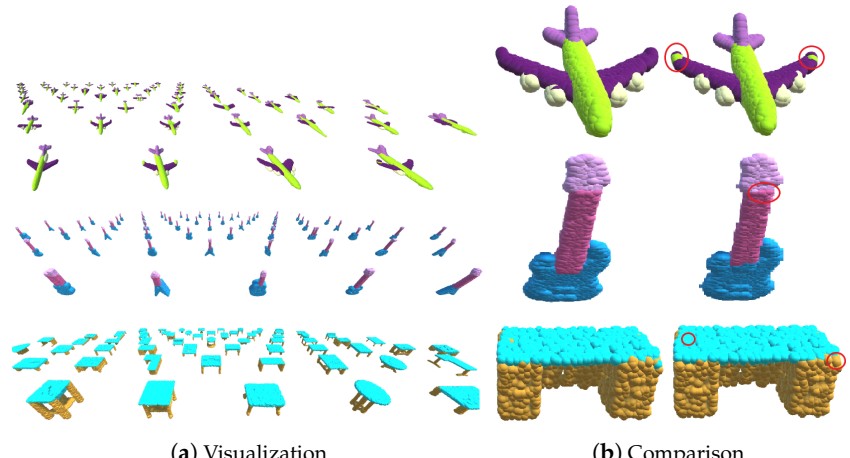

(**a**) Visualization                              (**b**) Comparison

**Figure 7.** Visualization of semantic part segmentation results. (**a**) visualizes some samples: airplane (top), guitar (middle), and table (bottom), while (**b**) visualizes the predictions and corresponding errors (right) compared to ground truth (left).

Tables 5 and 6 show the results on the S3DIS and KITTI dataset, respectively. It is worth noticing that our segmentation model is also efficient on a super large scale point cloud, which shows its great potential for autonomous driving applications.

**Table 5.** Segmentation results on S3DIS Area 5.

| Model | mIoU | Overall Accuracy |
|---|---|---|
| SegCloud [44] | 48.9% | - |
| PointNet [10] | 47.6% | 78.5% |
| DGCNN [12] | 56.1% | 84.1% |
| **OURS** | 50.8% | 85.0% |
| SPGraph [31] | **58.0%** | **86.4%** |

**Table 6.** Segmentation results on KITTI.

| Model | mIoU | Accuracy | Vehicle | Cyclist | Pedestrian | Background |
|---|---|---|---|---|---|---|
| SegCloud [44] | 36.8% | - | 67.5 | 7.3% | 53.5% | - |
| PointNet [10] | 38.1% | 92% | 76.7% | 2.9% | 6.6% | 89.8% |
| PCCN [45] | 58.1% | 95.5% | **91.8%** | 40.2% | 47.7% | 89.3% |
| OURS | **74.0%** | **98.1%** | 84.9% | **65.0%** | **61.3%** | **96.5%** |

## 5. Conclusions

In this paper, we propose a deep-wide neural network, named ShufflePointNet, to learn local representations for point cloud data. Experiments carried out show state-of-the-art performance for both the shape classification and semantic segmentation tasks on various datasets. The success of our model verifies that a deep-wide neural network can achieve high performance whilst reducing model complexity, which is crucial for real time applications.

In the future, we plan to apply our model on various real applications, such as environment perception for autonomous aerial or ground vehicles which typically need to process very large-scale point cloud data. In addition to geometric representations, we also plan to extend our model for other unstructured data (e.g., social networks). Furthermore, an efficient CNN-like operation development for unstructured data analysis is also a possible strategy to improve our model.

**Author Contributions:** Conceptualization, C.C.; methodology, C.C.; software, C.C.; validation, C.C. and L.Z.F.; formal analysis, C.C.; investigation, C.C. and L.Z.F.; resources, A.T.; data curation, C.C.; writing—original draft preparation, C.C.; writing—review and editing, C.C., L.Z.F., and A.T.; visualization, C.C.; supervision, L.Z.F. and A.T.; project administration, L.Z.F.; funding acquisition, A.T. All authors have read and agreed to the published version of the manuscript.

**Funding:** The HumanDrive project is a CCAV/Innovate UK funded R&D project (Project ref: 103283) led by Nissan and supported by world-class experts from nine other industry and academic organisations (Hitachi, Horiba MIRA, Aimsun, Atkins, University of Leeds, Cranfield University, Highways England, SBD Automotive Ltd. and the Transport Systems Catapult) is developing an advanced vehicle control system, designed to allow the vehicle to emulate a "natural" human driving style using machine learning and developing an Artificial Intelligence to enhance the user comfort, safety and experience. For more, check out www.humandrive.co.uk.

**Conflicts of Interest:** The authors declare no conflict of interest.

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
