# Peer review of "Go Wider: An Efficient Neural Network for Point Cloud Analysis via Group Convolutions"

_applsci, doi:10.3390/app10072391_

Round 1
Reviewer 1 Report
Paper is well written and introduce a new model for Neural network. However it seems, that very similar or almost equal paper was submitted or offered for by this team in September for Computer Science Computer Vision and Pattern Recognition. This paper or very similar is also available on ResearchGate as a preprint from September 2019. Please clarify these issues and explain the situation with the publication. If this paper was already, it is necessary to rewrite this paper completely and explain, where is a novelty.
There is also one question related to content. From the Web, the search seems, that ShufflePointNet is a relatively new solution, the first release on GitHub is from is in October and last contribution in Middle of November. It looks, like an extension of TensorFlow. Please can you in the text better clarify not only the methods but also implementation issue. Due to fact, that software is Open Source it could be useful for community.
Author Response
Point 1: Paper is well written and introduce a new model for Neural network. However it seems, that very similar or almost equal paper was submitted or offered for by this team in September for Computer Science Computer Vision and Pattern Recognition. This paper or very similar is also available on ResearchGate as a preprint from September 2019. Please clarify these issues and explain the situation with the publication. If this paper was already, it is necessary to rewrite this paper completely and explain, where is a novelty. 

Response 1: This paper is actually unpublished. In fact, the work was uploaded on Arxiv in September, and we then submitted to ICRA 2020 (International Conference on Robotics and Automation), but unfortunately we received a barely-rejection at the end of January, although we are quite confident with the performance and novelty of our ShufflePointNet.
Point 2: There is also one question related to content. From the Web, the search seems, that ShufflePointNet is a relatively new solution, the first release on GitHub is from is in October and last contribution in Middle of November. It looks, like an extension of TensorFlow. Please can you in the text better clarify not only the methods but also implementation issue. Due to fact, that software is Open Source it could be useful for community.
Response 2: We updated our code on Github with more details about implementation and compatibility issues, as most of issues are due to submodule installation problem and software version compatibility issues, so that our Github might be of better service to the community.

Reviewer 2 Report
This paper applies group convolutions to a neural network for efficient point cloud analysis. The structure is overall good. A few technical questions need to be clarified:
- The proposed algorithm seems to achieve comparable performance with PointCNN, please provide a more in-depth analysis of the comparison and the benefits of the proposed algorithm.
- Training details should include hardware and training time.
- The ablation study should include experiments to prove the usefulness of the channel shuffling operation.
Author Response
Point 1: The proposed algorithm seems to achieve comparable performance with PointCNN, please provide a more in-depth analysis of the comparison and the benefits of the proposed algorithm. 

Response 1: PointCNN and ShufflePointNet are point-based-convolution methods over unordered point cloud. PointCNN attempts to permute the neighboring points into a canonical order by learning a feature transformation, after which standard CNNs are possible to use. However, the performance is far from ideal because the position of the neighbors is unordered and unpredictable. On the other hand, ShufflePointNet uses MLP (Multi-Layer-Perceptron) convolutions over center points and their corresponding neighbors, which is naturally invariant to the point cloud data ordering. Besides this, the deep and wide architecture allows exploiting a sufficient number of features with a minor computational cost. However, the receptive field of CNN is much larger than that of MLP, which leads to the fact that PointCNN still can achieve a remarkable performance by leveraging the capabilities of CNNs over approximate ordering of neighbors. This has been better stated in the paper.
Point 2: Training details should include hardware and training time.
Response 2: We added the information about hardware and training time in the text. Our classification model was trained for 3 hours on a ThinkStation P920 workstation with 12 Intel E5-1650 V4 CPUs, 3.6GHZ, 32GB memory, 1 NVIDIA GTX 1080Ti. Our segmentation model is trained for 5 hours on a cluster server with four Intel E5-2620 v4 CPUs giving 32 CPU cores, 256GB of shared memory, and two NVIDIA TESLA V100 GPUs.
Point 3: The ablation study should include experiments to prove the usefulness of the channel shuffling operation.
Response 3: When we split feature channels into 2 groups and disable channel shuffling operation, the performance of overall accuracy drops significantly to 91.9%. We discuss that when we disable shuffling operation, center point features and corresponding neighboring points features are captured respectively, but still will concatenate before max pooling layer. As a result, without shuffling, only last MLP layer contributes to exploiting local features from both center points and corresponding neighboring points. This has been highlighted in the results section.
Reviewer 3 Report
- Please revise the writing quality, in terms of grammar errors and typos.
Typical mistakes include but are not limited to:
* incorrect use of (or lack of) definite and indefinite articles
* confusion of plural and singular forms (verbs, nouns)
* tenses
The following shows the examples.
Line 3: large amount of points -> a large amount of points
Line 200: have slightly drop -> have slightly droped
Line 234: all other settings follows -> all other settings follow
Line 245: Tables 6 and 5
Author Response
Point 1: Please revise the writing quality, in terms of grammar errors and typos.
Typical mistakes include but are not limited to:
* incorrect use of (or lack of) definite and indefinite articles
* confusion of plural and singular forms (verbs, nouns)
* tenses 

Response 1: Thank you for your comment. The paper has been proof checked and some sections completely rewritten.
Round 2
Reviewer 1 Report
The main concern about the paper was explained, but till now I have a number of concerns related to paper. Figure 1 and also Figure 2 are posted in text on places without relation to the text. In reality figure 1, which is very important for paper is not addressed in the paper at all.
There is no clear explanation of algorithms, without exact knowledge of other methods is difficult to understand all principles of ShufflePointNet.
Also, the architecture is explained only very generic, without better explanation of single components.
All article is described in not very clear form and it is not easy to read it.
There will be also good to describe better software implementatio.
Round 3
Reviewer 1 Report
The paper makes the significant process and I think, that it is possible to publish this paper.